# Frequency-Spectra-Based High Coding Capacity Chipless RFID Using an UWB-IR Approach

**DOI:** 10.3390/s21072525

**Published:** 2021-04-04

**Authors:** Kawther Mekki, Omrane Necibi, Hugo Dinis, Paulo Mendes, Ali Gharsallah

**Affiliations:** 1Laboratory for Research on Microwave Electronics, Physics Department, Faculty of Science, University of Tunis El Manar, 2092 El Manar, Tunisia; kawther.mekki@gmail.com (K.M.); ali.gharsallah@fst.utm.tn (A.G.); 2Computer Science Department, College of Arts and Sciences at Tabarjal, Jouf University, 72388 Jouf, Saudi Arabia; omrane.necibi@yahoo.com; 3CMEMS-Uminho, University of Minho, 4710-057 Braga, Portugal; hugodcdinis@gmail.com

**Keywords:** RFID, antenna, chipless tag, reader, UWB-IR, backscatter, amplitude

## Abstract

A novel methodology is proposed to reliably predict the resonant characteristics of a multipatch backscatter-based radio frequency identification (RFID) chipless tag. An ultra-wideband impulsion radio (UWB-IR)-based reader interrogates the chipless tag with a UWB pulse, and analyzes the obtained backscatter in the time domain. The RFID system consists of a radar cross-section (RCS)-based chipless tag containing a square microstrip patch antenna array in which the chipless tag is interrogated with a UWB pulse by an UWB-IR-based reader. The main components of the backscattered signal, the structural mode, and the antenna mode were identified and their spectral quality was evaluated. The study revealed that the antenna-mode backscatter includes signal carrying information, while the structural mode backscatter does not include any tag information. The simulation findings were confirmed by experimental measurements obtained in an anechoic chamber environment using a 6-bit multipatch chipless RFID tag. Finally, the novel technique does not use calibration tags and can freely orient tags with respect to the reader.

## 1. Introduction

Radio frequency identification (RFID) is a technique that has been applied to transmit and receive wireless data. It is intensively used in various applications such as automatic identification, asset tracking, and security surveillance [1,2]. To overcome the weaknesses of barcodes and other identification means such as the limitations of asset tracking, RFID technology has been employed in security access control, logistics, asset tracking, and supply chain management [3,4]. Traditional passive RFID systems use tags with no internal power source. These tags are powered by the electromagnetic energy transmitted from a RFID reader. However, the cost of RFID reduces its potential to replace trillions of barcodes [1,2]. To solve this problem, a chipless RFID tag is considered a good alternative [1]. The chipless RFID tag has no electronic circuitry and thus no intelligent signal processing capability. This option makes it less expensive and easier to mass produce at a unit cost comparable to optical barcodes [4,5,6,7,8].

RFID technology contains two major components: readers and tags [1,9,10]. Recently, the application of chipless tags has been extensively studied by the research community to develop high-performance RFID tags [3,11,12,13,14,15,16,17,18,19]. Several topologies relating to those used based on resonators have been suggested for chipless tags. Such topologies include circular loop [20,21,22], square loop [23,24,25], U-shaped [26,27,28], C-shaped [29,30], L-shaped [31,32,33,34], slotted [34,35,36], rhombic [37,38], octagonal [39], and microstrip-line [18,40,41]. It is also worth noting that the authors have suggested textile wearable application technology devices in the literature [42,43], which can be used as chipless RFID sensor tags in identification and tracking applications. There is also the three-dimensional chipless RFID tag suggested in [44]. Since the commercialization of chipless RFID requires a communication specification, it is important to consider the appropriate bandwidth for a frequency spectrum system. Therefore, the operating frequency spectra have been selected as 2–4 GHz [13,26,27], 2–5 GHz [14,29], 2–8 GHz [24,25], 3–6 GHz [32,34,38], 3–8 GHz [19,21,33], 5–8 GHz [16], 4–8 GHz [17], and 3.1–10.6 GHz [22,37]. The objective of this study was to enhance the performance of RFID tags in major design parameters such as reading range, tag and reader separation distance, bandwidth, bit encoding ability, and bit states/resonators. Since a communication protocol is needed for the commercialization of chipless RFID, it is necessary to establish an acceptable frequency range for high-capacity devices. Chipless RFID spectral-signature-based tags using a radar cross-section (RCS) have some apparent merits such as small size and they are easily readable.

We can distinguish between time domain (TD)-based tags and frequency domain (FD)-based tags. In the former, the interrogation of the tag by the RFID reader is carried out with a sequence of pulses. Subsequently, the tag code is determined based on the echoes reflected by a set of reflectors applied on the tag, which is linked to an object. More precisely, the existence or absence of these echoes and their time positions can be considered to specify the tag code [8,11,45,46,47]. On the other hand, in FD-based tags, an RFID reader uses a radio frequency (RF) signal to interrogate the tag. Afterward, the frequency signature is retransmitted by the tag to the reader. Each data bit shows the resonant frequency presence or absence in the operating spectrum [8,26,46,48,49,50].

Ultra-wideband (UWB) wireless communication technologies have recently attracted the attention of researchers due to their numerous benefits. Indeed, it enables lower power consumption, lower structural complexity, lower fabrication cost, size reduction, and a higher date rate. We would argue that (UWB) wireless communication technologies contain the majority of what researchers want to do in today’s technologies. For that, ultra-wideband (UWB) chipless tags for a RFID system may be a good solution for low-cost item tagging, as proven in [16,17,18].

The coding capacity of using UWB-impulse-based interrogation to remotely approximate the resonant features of a backscatter-based multipatch chipless RFID tag was enhanced in this study [51]. The use of ultra-wideband impulse radio (UWB-IR) based interrogation for the chipless RFID system has not been given much attention. However, a bit of research has been devoted to this field [52].

This paper presents an exploration of backscattering and its application to analyze the backscatter response created by an RCS-based RFID chipless tag for time and frequency domain analysis. The signal requests the tag when the response of a UWB signal is analyzed in the time and frequency domains in order to obtain the resonance information. The RCS and the advanced level signal processing of the backscattered signal play important roles in enhancing the operational range of chipless tags with enhanced tag design. To mitigate these challenges, the backscattered signals are recorded by a single antenna reader system using a standard frequency-domain backscattered tag with 6-bit data capability. A deeper insight into the mechanism of frequency-signature-based chipless RFID tags has been achieved through the time domain analysis of the tag backscatter. By time domain analysis, the valuable information-carrying component of the general backscattered signal was separated, and the frequency domain was then evaluated to estimate the tag’s frequency signature. In the CST Microwave studio suite, a condensed unit containing the reader antenna and the tag was simulated to gather the necessary simulated data. In both the time and frequency domains, different signal components were collected and assessed using the generated simulation data. The simulation results were also validated based on measurement findings provided by experiments carried out in an anechoic chamber environment. This multi-patch tag, which encodes 6-bit, has a large degree of freedom in the orientation angles with respect to the RFID reader antenna. Moreover, the maximum angular ranges can be detected from the measurement viewpoint. Therefore, the degradation of detection performance is quantified. The proposed system does not require calibration tags, and with respect to the reader location, the new approach produces appropriate results under various tag orientations and positions [4,19,34,35,53,54].

The rest of this manuscript is organized as follows. Section 2 presents the background on RCS-based chipless RFID tags; Section 3 describes the analyzed system model, the chipless RFID system, and the backscatter formed by the chipless RFID tag in which the responses is derived to the structural mode and the antenna mode backscatter; Section 4 describes the electromagnetic simulation environment as well as the experimental setup, which were used to obtain the simulation and measurement results; and Section 5 presents our concluding remarks.

## 2. Operating Principles of the Chipless Radio Frequency Identification (RFID) Reader System

The wireless data capturing method, namely RFID, uses RF waves to systematically identify objects. This technique employs RF waves to transmit data from the data carrying device, namely the RFID tag, to the reader [1,55,56].

### 2.1. Chipless Radio Frequency Identification (RFID) Reader System

The chipless radio frequency identification (RFID) system consists of a multiresonator chipless tag and an UWB reader antenna. The former system is a completely passive microwave circuit that employs spectral signatures to encode data. It is made of a multiresonator to accommodate multiple bits and operates over the UWB frequency spectrum. In fact, UWB antennas are generally operated to allow the interrogation signal to be forwarded by the reader and transfer the signal back to the reader after modulation of the frequency spectra by the multiresonator. In cases where the chipless RFID tag is hit by a vertically polarized electromagnetic (EM) wave produced by the transmitter antenna, a unique frequency signature is coded and the receiver antenna receives the encoded backscattered signal.

This unique frequency signature is then recorded and extracted by the RFID reader device, which enables the identification of the chipless RFID tag. The main function of the RFID reader is transmitting the interrogational signal toward the chipless tag, with magnitude and phase data. Subsequently, the interrogation signal is encoded into different frequency spectra according to the magnitude and phase and then retransmitted to the reader by the multiresonator chipless tag [7,21,49,57,58]. Figure 1 shows a block diagram of the introduced chipless RFID system.

### 2.2. Design of Ultra-Wideband (UWB) Reader Antenna

In our current work, we used circular monopole antennas, because, compared to other shapes with omnidirectional radiation characteristics and a simple structure, they have shown a much wider bandwidth. A radiating patch and ground plane on the same side of a dielectric substrate are the characterizing elements of an UWB monopole antenna.

The transmission and the reception of the interrogation signals by the reader were performed by employing a single monopole antenna. Figure 2a shows the antenna utilized in the experiment. Circular UWB monopole antennas have a simple design and a very wide bandwidth.

This antenna was a coplanar waveguide (CPW)-fed circular disk loaded monopole antenna [17,18,59]. The CPW feed line was separated from the ground by 0.25 mm. Due to its lower dielectric loss, decreased signal loss, and reduced thickness, RO4350B dielectric material is used to fabricate reader antennas at a lower cost. This allows us to miniaturize our proposed antennas. The Rogers RO4350B substrate had the following properties: εr = 3.48, δ = 0.0037, thickness of 0.76 mm, and copper cladding thickness of 175 µm.

Figure 2a shows the dimensions of the UWB monopole antenna and Figure 2b shows the fabricated antenna.

The reflection coefficients of the simulated and measured monopole antenna are shown in Figure 2c. The antenna performed from 2 to 7 GHz, and it had reflection coefficients of less than −10 dB.

### 2.3. Chipless Tag Antenna Design and Operation

The microstrip patch antennas used in this paper were designed to achieve narrow band operation, in contrast to the wide band antennas discussed earlier. This resonant behavior is desired in the design of frequency signature based chipless RFID tags. To encode tag-ID data as a frequency signature, we used sharp resonances, which is discussed later in Section 3.

To demonstrate the functionality of the system, an array containing six square microstrip patch antennas was used in the 6-bit chipless RFID tag [51].

The design of the multipatch-based chipless RFID tag is shown in Figure 3a. This tag was modeled on a RO4350B substrate material with a dielectric constant of 3.48, a loss tangent of 0.0019, a thickness of 0.76 mm, and a copper thickness of 175 μm. The lengths of the six patch antennas constituting the tag were 17.5, 16.5, 15.5, 14.5, 13.5, and 12 mm and they resonated at frequencies of 4.4, 4.7, 5.1, 5.4, 5.7, and 6.1 GHz.

Each patch antenna creates backscatter only at its corresponding resonant frequency, producing a distinctive frequency signature in the total backscattered signal. This resonant behavior has been proven to be efficient in designing chipless RFID tags based on the magnitude of the RCS [51]. Indeed, a RCS tag shows the tag-ID data-encoding frequency signature.

Six different ID tags were fabricated, and are illustrated in Figure 3b. The RCS was simulated and measured versus frequency and the peaks can be seen in Figure 3c.

## 3. Model of Chipless RFID System for Backscatter

In this section, we present the chipless RFID system model. The backscattering process is also described, and the major components of the backscattered signal are illustrated. In addition, the tag-ID data representation in the chipless RFID tag is shown, and the detection of the tag-ID data bits is investigated.

### 3.1. Chipless Tag Interrogation and Backscatter Response

The analysis of the backscatter signal from a multipatch-based chipless RFID tag was carried out in the time domain [51,56,60] and the overall system was described. Then, the system functioning, the chipless RFID tag design, and the nature of the backscatter were clarified based on the CST Microwave Studio simulation results. The applied chipless RFID system model is illustrated in Figure 4.

The circular monopole antenna utilized as the RFID reader represents an ultra-wide bandwidth system. In fact, the reader was employed to transmit the interrogation pulse and to receive the backscatter from the tag. The tag was located 30 cm in front of the reader antenna.

To interrogate the chipless tags, it was required to use a broadband pulse. The signal x(t) denotes the UWB pulse utilized to interrogate the chipless RFID tag. It represents a modulated Gaussian pulse with a 20 dB bandwidth of 6 GHz, whose frequency content varies between 2 and 8 GHz. This signal was applied to interrogate the chipless tags. The transmitted interrogation pulse, x(t), was obtained as follows [60]:
(1)$$X\left(t\right)\;={\;A}_{0}\;\cos\;\left(2{\pi \;f}_{c}\;t\right)\;\exp\;\left(\;-\frac{\left(t–\mu \right)2}{2\sigma^{2}}\right)$$

The mean and the standard deviation parameters specifying the form of the Gaussian pulse were *μ* = 0.6 ns and *σ* = 0.114 ns, respectively. The sinusoidal signal carrier amplitude and the frequency were A_0_ = 1 and f_c_ = 5 GHz, respectively. Figure 5a shows the time domain of the UWB interrogation pulse, while Figure 5b shows the frequency spectrum of the interrogation pulse.

As shown in Figure 4, x(t) is the UWB pulse transmitted by the RFID reader and y(t) denotes the overall received signal at the reader. In cases of interaction of the transmitted UWB pulse, x(t), with the tag, a portion of the UWB pulse will be captured by the individual patch antennas forming the tag, while the remaining part will be instantly reflected.

The total received signal, y(t), at the RFID reader is composed of the following three components:y(t) = y_r_ (t) + y_s_ (t) + y_a_ (t)(2)

The transmitted pulse rejection, y_r_(t), is considered as the largest and the first received component due to the antenna reflection coefficients.

At this point, the reader antenna fully transmits x(t) and receives any backscatter sent by the tag. The structural mode backscatter is the second received component, y_s_(t), whereas the weakest and the last received component is the antenna mode of the backscatter, y_a_(t).

### 3.2. Analysis of Chipless Tag Backscatter Response

Figure 6 illustrates the received signal of the full-time domain, y(t), provided by applying full-wave EM simulation in cases where the tag was located 30 cm away from the reader antenna.

First, a part of the signal yr(t), which represents the first and largest component of y(t), decreased progressively and disappeared. A small echo, which was the backscatter produced by the chipless tag, was noticed at 2.55 ns. Compared to the rejected signal, the backscatter was smaller and cannot be visibly observed in Figure 6. The structural mode backscatter and the antenna mode backscatter were clearly separated. Figure 6 also depicts the ys(t) and ya(t) components.

Here, the various components that formed the total signal arriving at the reader can be distinguished and separated.

The backscatter contained a large but short-duration initial component as well as a small but longer-duration component. The shape of the former resembles the transmitted Gaussian pulse represented in Figure 5b. We hypothesize that the structural mode backscatter, ys(t), was the larger component and that the antenna mode, ya(t), was the smaller component.

Nevertheless, as no transmission line presenting a controlled delay existed, as was the case in the chipless RFID tag depicted in Figure 6 before, we could not adequately determine the start of the antenna mode backscatter and the end of the structural mode backscatter.

To prove this hypothesis, it was necessary to separate the two components and investigate their spectral content.

The components of the received backscatter were isolated by employing a raised cosine window, as shown in Figure 7.

Figure 8 shows the spectral content of the windowed structural mode and windowed antenna mode provided by applying the fast Fourier transform (FFT) algorithm. The larger and first portion of the backscatter, ys(t), is characterized by a Gaussian amplitude spectrum resembling the spectrum of the transmitted UWB pulse (Figure 5b). It does not include any data on the resonant frequencies of the patch antennas in the chipless tag.

However, the amplitude spectrum of the secondary smaller component backscatter after the structural mode clearly displayed six spectral peaks, precisely at the resonant frequencies of the patch antennas in the chipless tag. The windowed ya(t) spectral content presents the resonant frequencies of the six individual patch antennas (f_1_ = 4.4, f_2_ = 4.7, f_3_ = 5.1, f_4_ = 5.4, f_5_ = 5.7, and f_6_ = 6.1 GHz).

Thus, the transient ya(t) after the initial strong backscatter ys(t) clearly contains the necessary information to approximate the resonant frequencies of the patch antennas in the chipless RFID tag. The backscatter antenna mode is viewed as a filtered variant of the tag’s UWB pulse incident, where only the frequencies associated with the patches are transmitted in, while the others are filtered out.

Therefore, the antenna mode backscatter spectral peaks showing the frequencies with higher energy (f_1_) in the incident UWB pulse were higher than those showing frequencies with lesser energy (f_2_) in the incident UWB pulse.

### 3.3. Tag Performance in Detection of Different Bits

The chipless tag presents 6-bit data. It is characterized by a single frequency spectral signature. It is possible to encode a data bit as 0 or 1 based on the peak or dip presence or absence in the spectrum. If a dip cannot be measured, the resulting bit will be low (i.e., ‘0’).

The frequency spectra of the antenna mode backscatter of the chipless RFID tags with various resonant patch antenna combinations were f_1_ = 4.4, f_2_ = 4.7, f_3_ = 5.1, f_5_ = 5.7, and f_6_ = 6.1 GHz. This proves that the presence of a resonant patch antenna in the chipless RFID tag provokes a corresponding peak in the chipless RFID tag spectral signature.

Figure 9 shows the frequency spectrum of the antenna mode backscatter provided to a chipless tag with only five resonators. It shows the disappearance of the fourth amplitude, f_4_, peak. This amplitude shows that the tag involved none of the f_4_ resonances. Therefore, it was possible to appropriately detect the tag-ID data in the tag as ‘111011’.

## 4. Experimental Validation in the Anechoic Chamber

### 4.1. Reader Antenna and Chipless RFID Tag Measurement Results

In this section, we discuss the experimental validation of the simulation results observed previously for the measurements obtained for the chipless RFID based on the fabricated multipatch UWB reader.

As shown in Figure 10, measurements were carried out in an anechoic chamber using a vector network analyzer (VNA). The tag and the reader antenna were mounted inside the anechoic chamber at a distance of 30 cm from each other and the reflection coefficient S_11_ and radiation pattern output measurements were performed.

The tag and the reader antenna placement inside the chamber are presented in Figure 10. The reflection coefficient profile, S_11_, and the radiation patterns were measured in the anechoic chamber. Figure 11a shows the reflection coefficient, S_11_, of the measuring antenna. The antenna performed, as expected, a reflection coefficient of less than −10 dB from 3.5 to 7.5 GHz. To validate the antenna radiation performance at the operating bands, the yz-plane (E-plane) radiation patterns of the chipless RFID tag system were measured as shown in Figure 11b.

For the chipless RFID tag, the radiation pattern was omnidirectional. The results provided by the radiation patterns did not change significantly over the entire functioning band.

This section explains how the RF Analog Signal Generator (E8663D PSG) and the Spectrum Analyzer (RSA306B) were used to measure normalized amplitude spectra in the frequency domain using the circulator and reader antenna to read the structural mode ys(t) and antenna mode backscatter ya(t). RF circulators play an important role in many RF communication systems, and for this measurement of the amplitude spectra in the chipless RFID tag, we used a RF circulator PE8432 from PASTERNACK. The RF circulator is a three-port system with a frequency range of 2–6 GHz, insertion loss of 0.8 dB, VSWR (voltage standing wave ratio) of 1.5, and isolation of >14 dB.

A reader with a single patch antenna was attached to a circulator on one side of port 2 for the transmission and reception of the signal. The signal is generated by a signal generator that is connected to port 1 of the circulator’s Tx transmission input on the other side. The signal reflected by the reader antenna is sampled at the reflected port attached to port 3 on the side of the circulator’s receive Rx output and seen on the spectrum analyzer.

A microwave circuit component was identified using a typical antenna for both transmitting and receiving functions with the use of a spectrum analyzer and signal generator. Figure 12 provides an example of how to calculate the amplitude spectrum of a chipless RFID tag.

Subsequently, backscatter from the chipless tag was received from these experimental frequency-domain measurements. Figure 13 displays the measurement results of the amplitude spectra of the ys(t) windowed structural mode backscatter as well as the ya(t) windowed antenna mode backscatter, and findings obtained from the electromagnetic simulations in which the chipless tag was placed in the measurement setup are shown in Figure 12 in front of the reader antenna. Agreement was observed between the measured results and the simulation findings.

### 4.2. Parametric Study of Orientation of Chipless Tag

The performance of the introduced method was validated experimentally. In our experiments, the tag was placed in various orientations relative to the reader antenna.

The rotation was carried out about the *z*-axis so that the patch antennas relating to the highest frequencies moved away from the reader antenna and the patch tag antennas relating to lower frequencies could approach the reader antenna.

As depicted in Figure 14a, the tag was placed opposite the reader antenna with the x, y, and z directions and rotation about the *z*-axis. Six resonant spectral peaks were observed for the various tag positions. The normalized amplitude spectrum results of the antenna mode backscatter for different label orientations are shown in Figure 14b. When the chipless tag rotated around the *z*-axis, the amplitude of the antenna mode frequency spectrum gradually decreased.

The correct frequency signature of the chipless RFID tag can be seen in Figure 14b. The normalized amplitude spectrum of the antenna mode does not undergo a major change when the rotation angles θ are 20 and 30°. Nevertheless, for rotations exceeding 45 and 60°, the normalized amplitude spectrum performance was reduced and some of the high resonant frequencies were not observed in the approximated frequency signature.

## 5. Conclusions

In this article, a new approach was proposed for the accurate estimation of the resonant features of a multipatch frequency-spectra-based chipless RFID tag. The UWB-IR-based reader architecture was used to query the tag where the backscatter was evaluated in the time and frequency domains. It was found that the information-carrying portion of the obtained signal was contained in the antenna mode of the backscatter, whereas the structural mode of the backscatter does not include information about the resonant characteristics of the chipless tag. The results obtained from the simulation of a RFID system that consists of a chipless tag containing a six square array of microstrip patch antenna in which a UWB-IR based reader interrogates the chipless tag with a UWB pulse were validated by the measurement results from experiments performed in an anechoic chamber environment. The proposed approach is capable of effectively estimating the resonant features of a chipless tag without the use of calibration tags, and additional signal processing up to a distance of 30 cm. The implications of the direction and position of the tag with respect to the reader on the calculation of the tag frequency spectra were also studied. It was shown that for tag rotations of less than 45°, the approximate frequency spectra can be used to detect the information found in the chipless tag. This study sheds light on the understanding of the backscattered signal in the frequency domain from a 6-bit multipatch chipless RFID tag. This is a proposal to leverage the development of realistic solutions to use in applications such as authentication and security of chipless RFID technology.

## Figures and Tables

**Figure 1 sensors-21-02525-f001:**
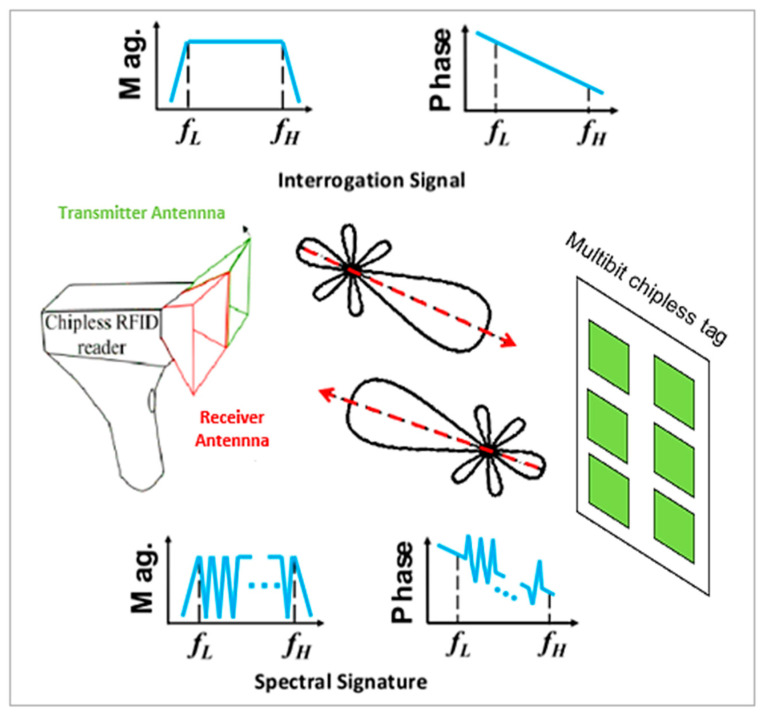
Operation principles of the chipless radio frequency identification (RFID) system.

**Figure 2 sensors-21-02525-f002:**
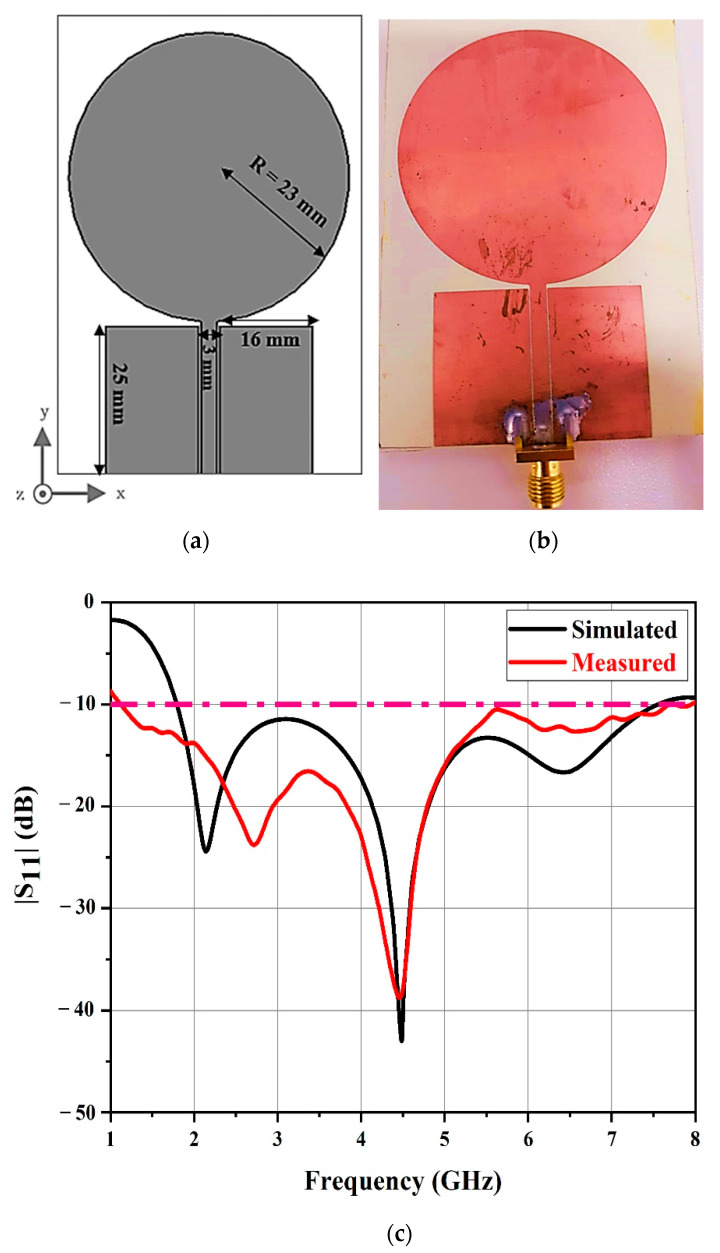
Dimensions of (**a**) the reader antenna and (**b**) the fabricated reader antenna. (**c**) Simulated and measured reflection coefficients of the reader antenna.

**Figure 3 sensors-21-02525-f003:**
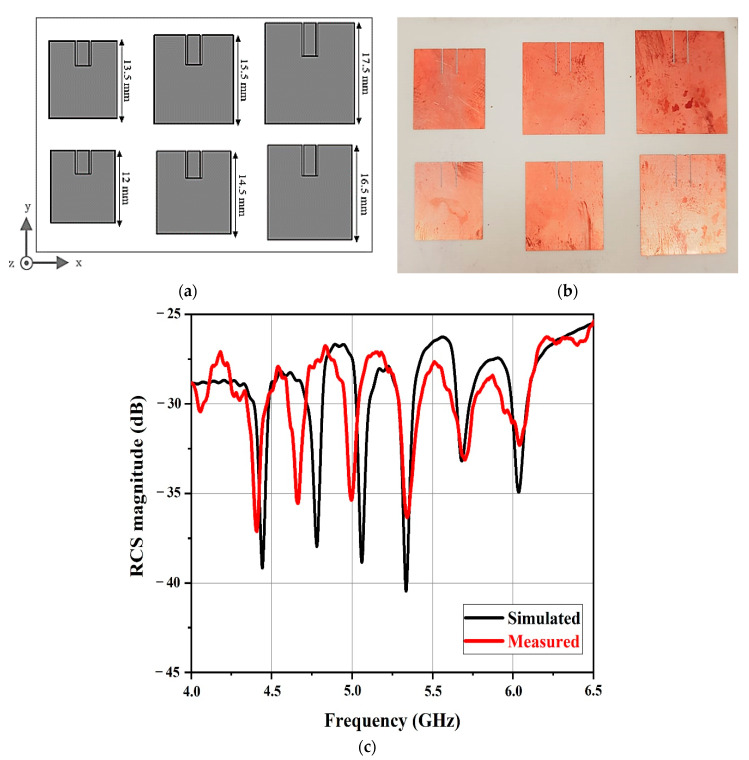
Dimensions of (**a**) the chipless tags loaded with resonators and (**b**) the fabricated chipless tags. (**c**) The simulated and measured radar cross section (RCS) magnitude versus frequency.

**Figure 4 sensors-21-02525-f004:**
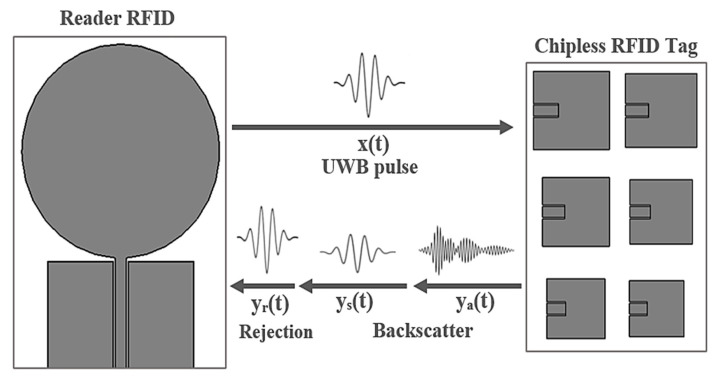
System model for the multipatch-based chipless RFID reader.

**Figure 5 sensors-21-02525-f005:**
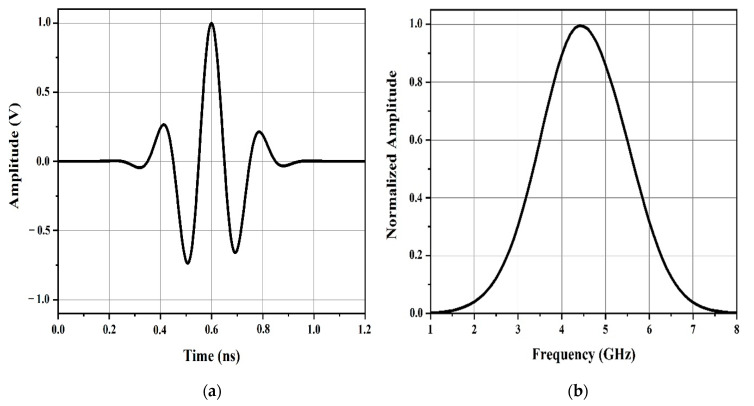
(**a**) Time domain of the ultra-wideband (UWB) interrogation pulse and (**b**) frequency spectrum of the interrogation pulse.

**Figure 6 sensors-21-02525-f006:**
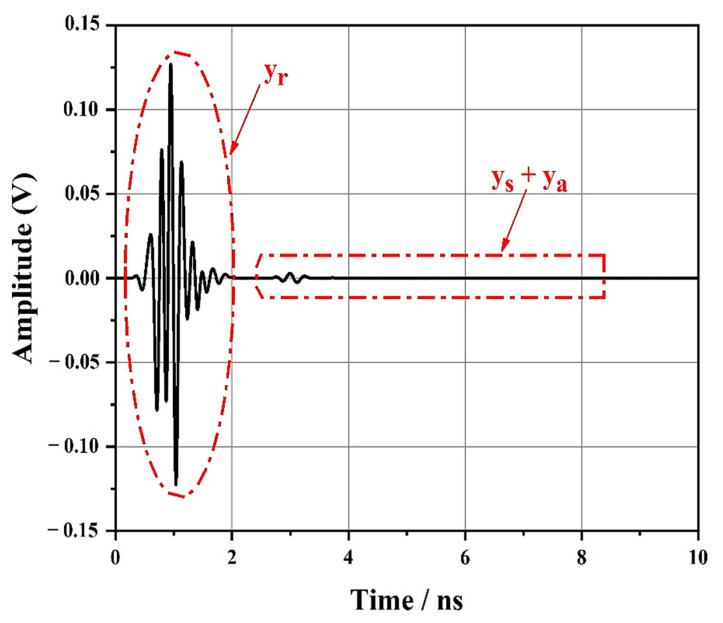
Total received signal, y(t) at the antenna.

**Figure 7 sensors-21-02525-f007:**
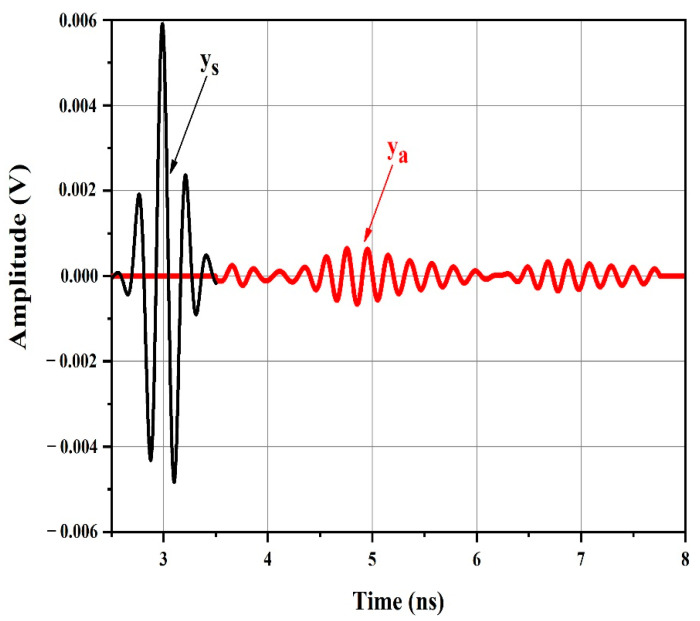
Magnified portion showing the structural mode ys(t) and antenna mode backscatter ya(t).

**Figure 8 sensors-21-02525-f008:**
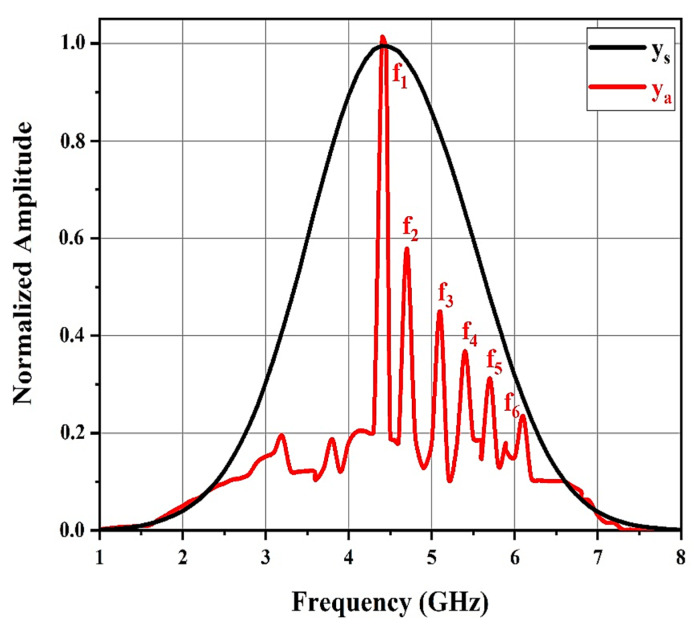
Normalized amplitude spectra of the structural mode ys(t) and antenna mode backscatter ya(t) obtained using fast Fourier transform (FFT).

**Figure 9 sensors-21-02525-f009:**
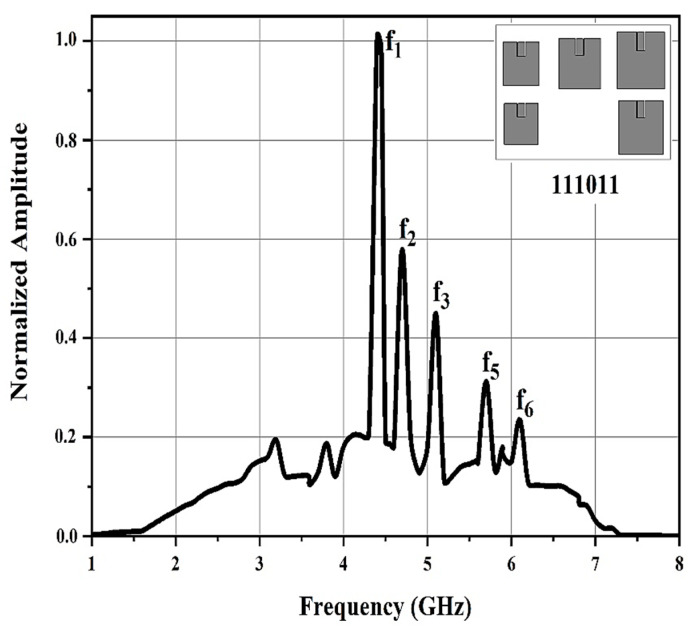
Frequency spectrum of the antenna mode backscatter for the chipless RFID tag containing data ‘111011’.

**Figure 10 sensors-21-02525-f010:**
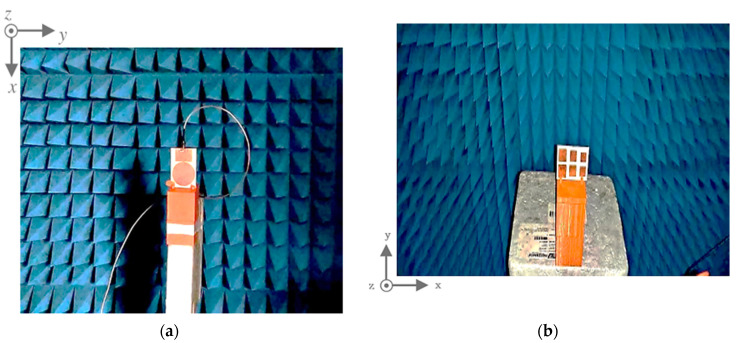
Experimental setup in an anechoic chamber with (**a**) the reader antenna and (**b**) the chipless tag, on their supports.

**Figure 11 sensors-21-02525-f011:**
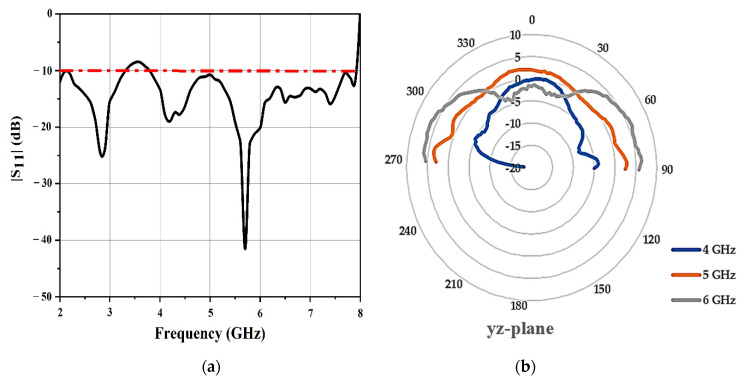
(**a**) Measured reflection coefficients of the antenna and (**b**) measured yz-plane (E-plane) radiation pattern at different frequencies.

**Figure 12 sensors-21-02525-f012:**
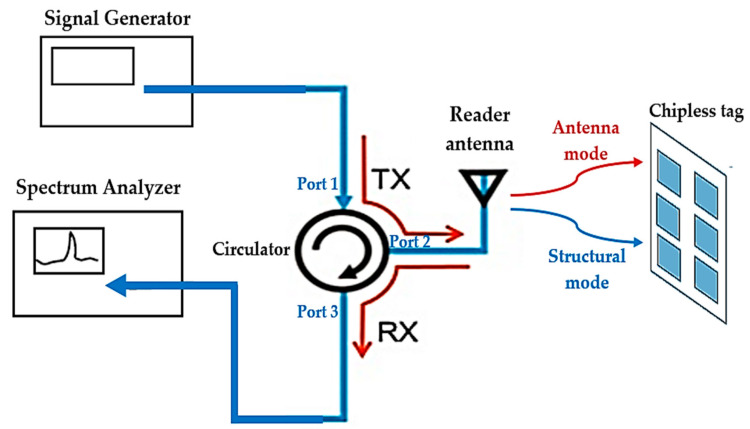
The reader antenna connected to a circulator with a spectrum analyzer and a signal generator to receive the amplitude spectra of the structural mode and the antenna mode in the chipless tag.

**Figure 13 sensors-21-02525-f013:**
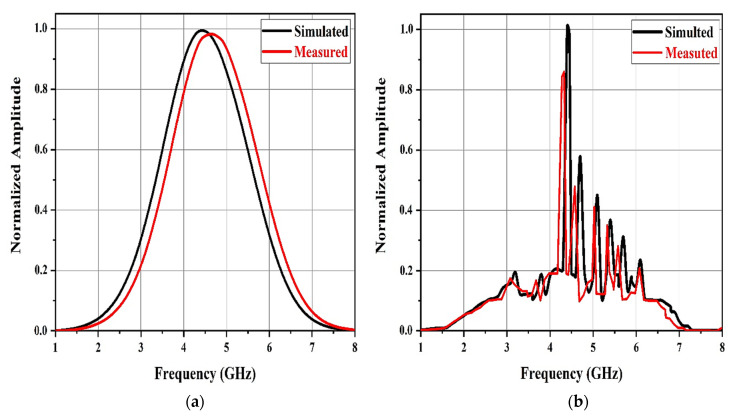
Measured normalized amplitude spectra of (**a**) the structural mode ys(t) and (**b**) the antenna mode backscatter ya(t).

**Figure 14 sensors-21-02525-f014:**
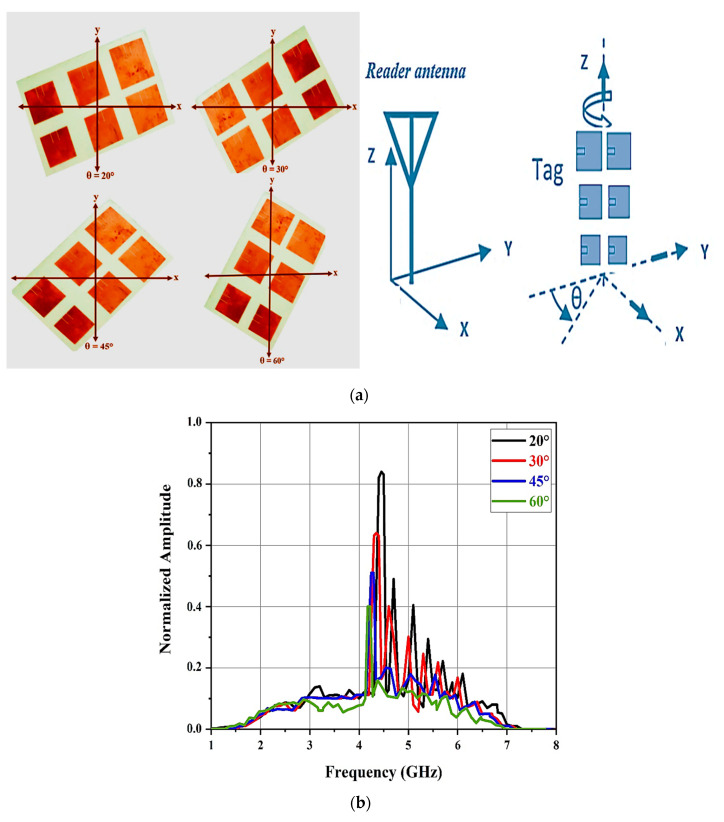
(**a**) Position of the tag with respect to the RFID reader and (**b**) normalized amplitude spectrum of the antenna mode backscatter for different tag orientations.

## Data Availability

Data sharing not applicable.

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
