# Peer review of "Frequency-Spectra-Based High Coding Capacity Chipless RFID Using an UWB-IR Approach"

_sensors, 2021, doi:10.3390/s21072525_

Round 1

Reviewer 1 Report

This manuscript proposes a chip-less RFID tag. Globally, the work is novel and the results are inovative. I like the way the experiments are described and I like the results. I have no doubts to recommend this work for publication on these transaction after minor amendments.

My remarks are the following:

(1) please indicate the coordinate system associated with figure 10, and place in the same figure

(2) the coordinate system must also be associated with the radiation diagram of figure 11 b

(3) regarding fig 12, please indicate the technical characteristics of the circulator and the characteristics of the spectrum analyzer used in the measurements.

(4) if possible, specify the manufacturer and the model of RF circulator.

(5) there was any special reason for using the Rogers RO4350B RF substrate, in comparison with other substrates?

Reviewer 2 Report

In general, this paper is well written. However, the following two concerns need to be addressed before publication:

  1. The introduction section needs to be reorganized, as it currently consists of too many paragraphs, making it harder to capature the main contributions of the work.
  2. Several sentences are too long, e.g. “The traditional passive RFID consists of a tag that has an 36 antenna that, when interrogated by an RFID reader, harvests energy from the signals to power an 37 integrated electronic circuit that assists in the process of wireless communication.”
    Please break it into two or more sentences to enhance the readability.

Reviewer 3 Report

The authors proposed a novel methodology  to reliably predict the resonant characteristics of a multipatch backscatter-based radio frequency identification (RFID) chipless tag. 

Significant: Yes, the paper is a significant advance or contribution.

Supported: Mostly yes,

Referencing: some additions are necessary

Quality: The organization of the manuscript and presentation of the data and results need some improvement.

Data: Yes, but some results are necessary

Whilst the paper shows promising initial results. A key theme includes the need to focus on the novel contributions and limit the explanation of well-known methods and definitions in the field. The paper is not clear. I also recommend major reorganisation of the paper.

I suggest the authors the following papers for reading:

 https://doi.org/10.3390/electronics10040478 
 https://doi.org/10.3390/electronics9122116
 https://doi.org/10.3390/iot1010007
 https://doi.org/10.3390/s20174740
 https://doi.org/10.3390/s20071843 
 https://doi.org/10.3390/s20020429
 https://doi.org/10.3390/s19214785
 https://doi.org/10.3390/electronics5040077
 https://doi.org/10.3390/s19030494 

Round 2

Reviewer 3 Report

The paper has been improved, but I suggest the authors increase the introduction , the section results, and check the references, recommended in the first review. The quality of the work has improved a lot.
